# Algae-Bacteria Consortia as a Strategy to Enhance H_2_ Production

**DOI:** 10.3390/cells9061353

**Published:** 2020-05-29

**Authors:** Neda Fakhimi, David Gonzalez-Ballester, Emilio Fernández, Aurora Galván, Alexandra Dubini

**Affiliations:** Departamento de Bioquímica y Biología Molecular, Facultad de Ciencias, Universidad de Córdoba, Campus de Rabanales, Edif. Severo Ochoa, 14071 Córdoba, Spain; z72fafan@uco.es (N.F.); dgballester@uco.es (D.G.-B.); bb1feree@uco.es (E.F.); bb1gacea@uco.es (A.G.)

**Keywords:** algae, bacteria, biohydrogen, *Chlamydomonas reinhardtii*, co-cultures, consortia, hydrogen

## Abstract

Biological hydrogen production by microalgae is a potential sustainable, renewable and clean source of energy. However, many barriers limiting photohydrogen production in these microorganisms remain unsolved. In order to explore this potential and make biohydrogen industrially affordable, the unicellular microalga *Chlamydomonas reinhardtii* is used as a model system to solve barriers and identify new approaches that can improve hydrogen production. Recently, Chlamydomonas–bacteria consortia have opened a new window to improve biohydrogen production. In this study, we review the different consortia that have been successfully employed and analyze the factors that could be behind the improved H_2_ production.

## 1. Introduction

Finding renewable, sustainable and clean energy sources has become one of the main priorities of our society. Hydrogen (H_2_) is a promising clean and carbon-free energy source with a high energy value (142 kJ/g) that can be easily interconverted with electricity and used for domestic and industrial applications. Currently, H_2_ production techniques include steam reforming natural gas/oil, coal gasification, biomass gasification/pyrolysis, and electrolysis and thermolysis of water. All these techniques are either polluting and/or demand a large amount of energy [1,2]. Under this scenario, the biological production of H_2_ (bioH_2_) has garnered considerable attention in recent decades, as it could be a cheap and renewable source of fuel. Different microorganisms such as microalgae, cyanobacteria, photosynthetic bacteria and some heterotrophic bacteria can produce H_2_ [3,4]. Algae and cyanobacteria are well-known photoautotrophic organisms able to convert CO_2_ into organic matter and release O_2_ during this process. Under specific conditions, H_2_ production is linked to photosynthetic activity. Non-oxygenic photosynthetic bacteria can also use light and organic acids (and other chemical forms) to obtain energy and produce H_2_, without releasing O_2_. Heterotrophic bacteria, on the other hand, can degrade organic matter and release CO_2_, with some of them also producing H_2_. Among them, photobiological H_2_ evolution by green algae and cyanobacteria has attracted considerable attention since, potentially, they do not require organic carbon sources to produce H_2_, only water and sunlight [4,5,6]. Moreover, microalgae and cyanobacteria are the most dominant photosynthetic organisms on Earth, which increases their biotechnological interest. However, photosynthetic H_2_ production is still inefficient for industrial implementation due to its low yield and rate of H_2_ generation. One of the most important bottlenecks of biological H_2_ production is its sensitivity to oxygen (O_2_). In all the H_2_-producing microorganisms, O_2_ is a strong repressor of H_2_ production. 

### 1.1. H_2_ Production in Green Algae 

*Chlamydomonas reinhardtii* (Chlamydomonas throughout) is a unicellular green microalga able to grow autotrophically and heterotrophically that has been chosen as a model system to study H_2_ photoproduction. There are three different pathways that can lead to H_2_ production in Chlamydomonas. Two of them are linked to the photosynthetic electron chain, while a third is linked to fermentative metabolism. In the photosystem II (PSII)-dependent pathway (also termed the direct pathway), the electrons generated at the level of PSII from water splitting are transferred to the photosynthetic electron chain, where they ultimately reach photosystem I (PSI) and the ferredoxins (FDXs), which are the final electron donors to the hydrogenases (HYDAs) [7,8]. Since this pathway require the activity of the PSII, both electrons and O_2_ are simultaneously generated. In the PSII-independent (or indirect) pathway, NAD(P)H acts as a source of electrons that can directly reduce the cytochrome b_6_f through type II-NADH dehydrogenase (NDA2) [9,10]. Once the electrons are in the photosynthetic electron chain, they reach the PSI and the FDXs as in the PSII-dependent pathway, but in this case O_2_ is not co-generated with H_2_ since PSII does not participate in the generation of electrons [11]. In the PSII-independent pathway, starch degradation has been identified as the most common source of reductants under sulfur (S)-depleted conditions [12]. However, under hypoxia and nutrient replete conditions, acetic acid assimilation has been suggested to play an important role as source of reductants for H_2_ production [13,14,15,16]. The third pathway is known as the fermentative or dark pathway. Here, the Pyruvate Ferredoxin Oxidoreductase (PFR) enzyme oxidizes pyruvate to acetyl CoA under anoxic conditions. This reaction is coupled with the generation of electrons, which are transferred to the HYDAs via FDXs [8,17,18]. In Chlamydomonas, the dark H_2_ production is quantitatively much more reduced than H_2_ photoproduction.

As mentioned before, the main drawback of photohydrogen production in algae is caused by the O_2_ sensitivity of the HYDAs, which show inhibitory effects at both transcriptional and posttranslational levels [19,20]. Therefore, H_2_ photoproduction in green algae occurs under anoxic/hypoxic conditions and, at a physiological level, H_2_ production is a transitory phenomenon since O_2_ and H_2_ co-evolution are incompatible. This is especially true for the PSII-dependent pathway. Furthermore, the process encounters several other bottlenecks that decrease the efficiency of H_2_ evolution. Among them are low light conversion efficiency, the non-dissipation of the proton gradient, the competition between electron acceptors for photosynthetic electrons, the reversibility of the HYDAs, the low level of HYDAs expression, and the pH inhibition (reviewed in [21,22,23,24,25]). Several genetic modifications have successfully palliated some of these limitations [23,25,26]. Different culturing approaches have also been developed to alleviate the identified bottlenecks. Among these approaches are the modulation of the light intensity [14,27,28,29,30,31], the optimization of the photosynthetic electrons flow towards the HYDAs [29,32,33,34], the implementation of nutrient stresses, especially sulfur (S) deprivation, influencing H_2_ production [35,36,37,38,39], the addition of O_2_ scavengers into the culture media [33,40,41], or cell immobilization [42,43,44]. Moreover, in recent years, the co-cultivation of alga and bacteria has arisen as an alternative strategy to increase algal H_2_ production. 

### 1.2. H_2_ Production in Cyanobacteria 

Cyanobacteria are prokaryotic photosynthetic microorganisms able to grow heterotrophically or photoautotrophically, some of which are nitrogen-fixing. During phototrophic growth, they perform oxygenic photosynthesis using an electron transport chain similar to algae and plants. Like in microalgae, H_2_ production through the HYDAs can be linked to the photosynthetic activity or to the fermentative pathways. However, unlike microalgae, H_2_ production can also be linked to the N_2_ fixation mediated by the nitrogenases. Both HYDAs and nitrogenases are O_2_ sensitive. Among cyanobacteria, the best H_2_ producers link H_2_ production to nitrogenase activity, since cyanobacteria HYDAs are highly reversible, and their most common physiological role is related to H_2_ uptake. Nitrogenases are only expressed under nitrogen-limiting conditions, and nitrogenase-based H_2_ production is very expensive in terms of energy expenditure (e.g., 15 photons/H_2_ are required by nitrogenases vs. four photons/H_2_ by HYDAs) [8,45].

### 1.3. H_2_ Production in Non-Oxygenic Photosynthetic Bacteria

Some non-oxygenic photosynthetic bacteria can also produce H_2_. In this group of microorganisms, the Purple Non-Sulfur Photosynthetic (PNSP) bacteria are among the best known H_2_ producers. As with cyanobacteria, H_2_ production by PNSP bacteria is mostly linked to nitrogenase activity. ATP generated during photosynthesis is used by the nitrogenases to produce NH_3_ and H_2_. In this case, photosynthesis is not linked to water splitting and thereby O_2_ is not produced. Instead, the most common source of electron donors are organic acids, and the process is known as photo-fermentation. For H_2_ production, formate, acetate, lactate and butyrate can act as electron donors, with butyrate being the best inducer of H_2_ production [8,46,47]. Like cyanobacteria, two of the main factors limiting H_2_ production in PNSP bacteria are the simultaneous occurrence of H_2_ uptake and the need to establish nitrogen-deficient conditions. 

### 1.4. H_2_ Production in Heterotrophic Bacteria

Many heterotrophic bacteria can produce H_2_ though fermentative pathways (also known as dark H_2_ production). Bacterial fermentation of sugars can produce a large variety of fermentative end products, including H_2_. There are two distinctive groups of bacteria that have been extensively studied regarding fermentative H_2_ production. One group is composed of strict anaerobes (represented by, e.g., *Clostridium* spp.), where H_2_ production is linked to the oxidation of pyruvate into acetyl CoA by Pyruvate Ferredoxin Oxidoreductase (PFOR). This pathway is known as the PFOR pathway. The second group are facultative anaerobes (represented by, e.g., *E. coli*), which, under anaerobic conditions, perform so-called mixed acid fermentation, where pyruvate can be used by Pyruvate Formate Lyase (PFL) to produces formate and acetyl CoA. Formate is then converted to CO_2_ and H_2_ by the Formate Hydrogen Lyase (FHL), and the process is known as the PFL H_2_-production pathway. H_2_ production through dark fermentation has several limiting factors, including 1) the existence of other competitive fermentation pathways and, 2) the excessive accumulation of end products (mainly ethanol, formate, acetate, lactate, succinate, glycerol and butyrate) that block H_2_ production [48,49,50,51,52,53].

Although numerous efforts have been made to improve H_2_ production in algal and bacterial systems, the integration of these two systems to improve bioH_2_ production has received less attention [8,21,47]. This review outlines the past and recent achievements obtained when the green algae Chlamydomonas is co-cultivated with different bacterial strains to improve H_2_ production.

## 2. Current Achievements Obtained with Chlamydomonas-Bacteria Consortia

Several studies have proven the possibility to improve H_2_ production when using co-cultures of alga and bacteria [21,54,55], with some of them focusing on the use of the alga Chlamydomonas. 

Table 1 provides a comparative analysis of all the previously published data about H_2_ production in Chlamydomonas–bacteria consortia with their respective algal monocultures in terms of yield, rate and sustainability. Studies are ranked according to the total H_2_ production yield. Notably, most of the publications show enhancements in H_2_ production parameters (yield, rate and duration) in the co-cultures relative to the monocultures, with many consortia promoting a threefold yield enhancement. Different *Pseudomonas* sp. and *Bradyrhizobium japonicum* are bacterial partners that lead to the highest H_2_ production yields in cultures incubated in Tris-Acetate-Phosphate (TAP) medium, devoid of S (TAP-S), and they often lead to great enhancements in H_2_ production (up to 22.7-fold and 32.3-fold, respectively) (Table 1). Note that these two bacterial partners are not known to be H_2_ producers by themselves. In general, the best condition for H_2_ production can be obtained in TAP-S (Table 1), confirming that, as in the case of Chlamydomonas monocultures, S deprivation is a physiological condition that greatly promotes H_2_ production in this alga. The light intensity does not seem to be a crucial parameter for H_2_ production from consortia incubated in TAP-S (Table 1). H_2_ photoproduction in Chlamydomonas monocultures in TAP medium is scarce, unless low light intensities (below 22 PPFD) are used [14]. However, different consortia can attain noticeable H_2_ production in TAP medium at higher light intensities (Table 1), which open the possibility to further explore H_2_ production under non-stressful conditions to avoid S removal and two-phase bioreactors. Finally, the use of H_2_-producing bacterial strains such as wild-type strains of *E. coli* in media supplemented with glucose brings up the possibility to combine H_2_ production from both alga and bacterium [56]. This consortium can produce up to 32.7 mL/L, which is higher than the production reported for other consortia in TAP-S medium (Table 1). Similarly, other bacteria like *Pseudomonas putida* and *Rhizobium etli* can also facilitate H_2_ production in Chlamydomonas when incubated with sugars as the only carbon sources (Table 1).

To contextualize the achievements obtained using Chlamydomonas–bacteria consortia, Table 2 lists some of the most successful strategies described in Chlamydomonas for H_2_ production, including monocultures and co-cultures, and ranks them by the total H_2_ yield obtained. Monocultures using genetically modified strains and S deprivation can lead to the highest H_2_ yields. However, the use of Chlamydomonas wild-type strains co-cultured with different bacterial partners under S deprivation are also ranked within the top list. For example, different co-cultures incubated in TAP-S employing *Pseudomonas* sp. or *Bradirizhobium japonicum* have achieved ≈165–170 mL H_2_/L culture [57,58,59], and there is a published patent for H_2_ production using Chlamydomonas and *Pseudomonas fluorescens* co-cultures claiming to produce 196 mL/L [68]. These values obtained using co-cultures are a bit far from the maximal Chlamydomonas H_2_ production reported (850 mL/L) using a proton gradient mutant (*pgrl5*) affecting the cyclic electron transfer [69]. However, co-culturing techniques could have a great potential to further improve H_2_ production if genetically modified Chlamydomonas (or bacterial) strains are employed in co-cultures. Moreover, it should be noted that most studies exploring H_2_ production in Chlamydomonas co-cultures are very recent and there are much more possibilities to explore in this field.

## 3. Characteristics of the Algae–Bacteria Association for H_2_ Production

In recent years, an increased interest in the study of algal–bacterial interactions has emerged not only due to their ecological significance, but also for their biotechnological potential. It is known that algae and bacteria affect one another’s physiology and metabolism. In natural ecosystems, algal–bacterial interactions cover a whole range of relationships: mutualism, commensalism and parasitism, depending on specific species and living requirements [77]. These interactions are omnipresent in all ecosystems. Moreover, microorganisms have complex and very versatile metabolisms, allowing them to grow or to simply survive in non-optimal environments. In this sense, Chlamydomonas, for example, apart from its photoautotrophic metabolism, has a fermentative metabolism that allows this alga to consume internal reserves such as starch under anaerobic conditions, releasing H_2_ and other end products to the medium. Moreover, Chlamydomonas can also grow heterotrophically and is able to consume acetic acid as a carbon source. Noticeably, acetic acid is the only organic carbon form that Chlamydomonas can uptake and, under hypoxic conditions, it has also been suggested that the assimilation of this compound is connected to H_2_ production in this alga [14,15,16]. 

At a physiological level, the production of H_2_ by microorganisms is considered as an escape valve for the electrons generated in excess during either photosynthetic or fermentative processes. The activation of hydrogenases (or nitrogenases) occurs under very specific environmental conditions, and for most microorganisms, H_2_ production can be considered as a transitory event. When cultivating axenic cultures of H_2_-producing microorganisms in the laboratory, different growth conditions are used to maximize H_2_ production. However, the complex interplay between the different microorganisms has not been not studied. Understanding this interplay can provide valuable information to overcome some of the bottlenecks associated with biological H_2_ production. 

A straightforward advantage of co-culturing heterotrophic bacteria with algae is that they can efficiently remove O_2_ from the media, which is the most critical bottleneck associated with H_2_ photoproduction. At the same time, the CO_2_ released during bacterial fermentation can support algae and cyanobacteria growth, while the photosynthetic O_2_ production can support the growth of facultative anaerobic bacteria. In addition, algae and photosynthetic bacteria can theoretically combine their sunlight wavelength absorption ranges to increase the overall light-to-energy conversion efficiency for H_2_ production or for biomass generation. Finally, several photosynthetic and fermentative metabolites can be exchanged between microorganisms, establishing specific nutrient fluxes that can benefit H_2_ production and/or growth. Among these nutrient fluxes, carbon fluxes are quantitatively the most prominent, although nitrogen, phosphorous and S sources, and growth factors like Vitamin B12, have also been reported as favoring algae–bacteria interactions [78,79,80,81,82].

In the following sections, the potential mechanisms influencing H_2_ production in algae–bacteria cultures are discussed. They are categorized according to the impact on (1) biomass, accumulation of internal reserves, and metabolite exchange supporting H_2_ production, (2) net O_2_ evolution, and (3) the possibility to extend the solar spectrum absorption range. 

### 3.1. Biomass, Accumulation of Internal Reserves and Metabolite Exchange Supporting H_2_ Production

#### 3.1.1. Starch Accumulation could be Promoted in Co-Cultures

Starch reserves in Chlamydomonas can be connected to photobiological H_2_ production through the PSII-independent pathway (Figure 1). This pathway relies on the non-photochemical reduction in the PlastoQuinone (PQ) pool using the electrons derived from NAD(P)H [9,10,11,83]. The glycolytic degradation of starch is proposed to be the main source of electrons for this H_2_-producing pathway during S deprivation conditions [12]. Moreover, starch degradation can also feed the fermentative or dark H_2_ production in Chlamydomonas via the PFR pathway (Figure 1) [8,17,18]. Different nutrient stresses (mainly N and S) can promote starch accumulation in Chlamydomonas cultures under both light and dark conditions [84,85], which, in turn, can favor H_2_ production.

Recently, it has been observed that co-culturing Chlamydomonas with different bacterial strains can lead to high starch accumulation in this alga. These bacterial strains include *Bradyrhizobium japonicum* [58], *Azotobacter chroococcum* [67], *Pseudomonas* sp. [59] and *Thuomonas intermedia* [65]. However, the precise reasons explaining why the starch accumulation occurs in these co-cultures have not been elucidated. In any case, co-culturing Chlamydomonas with certain bacterial strains could be used as an approach to promote starch accumulation, which potentially can enhance algal H_2_ production through the PSII-independent pathway or through metabolite exchange (see below sections) (Figure 1). 

#### 3.1.2. Mobilization of the Algal Starch Reserves Can Provide Organic Acids for H_2_ Producing Bacteria

Chlamydomonas has a very versatile fermentative metabolism and is able to quickly degrade starch reserves under anaerobic conditions to different fermentative end products including H_2_ [86,87,88] (Figure 1). Some end products are secreted to the medium by wild-type Chlamydomonas cultures, including acetate, ethanol and formate. Glycerol, succinate and lactate are minor end products secreted by most wild-type Chlamydomonas cells; however, the noticeable excretion of these fermentative products can be found in some Chlamydomonas mutants [86] or in some strains considered to be wild-type [89]. All these secreted end products can be theoretically used by bacteria as electron donors for H_2_ production (Figure 2), and some of them have been probed at an empirical level using Chlamydomonas–PNSP bacteria cultures [62,90]. Miyamoto et al. [62] reported that, when co-culturing Chlamydomonas and *Rhodospirillum rubrum*, they both produced H_2_ in dark conditions. In the case of Chlamydomonas, H_2_ originated from the fermentative degradation of the starch reserves, while, in the case of *R. rubrum*, H_2_ originated from the Formate Hydrogen Lyase pathway using the formate excreted by the alga as a substrate. Similarly, Miura et al. [90] reported that after incubating Chlamydomonas in the dark, the resulting medium broth was used by a marine photosynthetic bacterium, *Rhodopseudomonas* sp., to photoproduce H_2_. This Chlamydomonas medium broth was enriched with acetic acid and ethanol. 

#### 3.1.3. Acetic Acid Exchange Can Promote H_2_ Production in both Algae and Bacteria

As mentioned before, the donation of fermentative metabolites from Chlamydomonas to different bacteria can promote bacterial H_2_ production. However, the opposite flux (from bacteria to alga) can also benefit both algal and bacterial H_2_ production, especially when acetic acid is produced and secreted by the bacteria [56]. In Figure 2, some of the metabolites that can be potentially exchanged between algae and other microorganisms during both growth and H_2_ production conditions are depicted.

Many bacteria can produce H_2_ though fermentative pathways (dark H_2_ production). In organisms using the PFOR H_2_-production pathway (e.g., *Clostridium* spp.), the highest yield is obtained when acetate is the main fermentation end product. Similarly, in organisms using the PFL H_2_-production pathway (e.g., *E. coli*), the highest yield is obtained when acetic acid and ethanol are the end products (Figure 2) [48,49]. The maximum theoretical yield of dark H_2_ production is assumed to be 2 to 4 mol of H_2_ per mol of glucose, depending on the kind of microorganisms (2 moles for facultative aerobes and 4 moles for strict anaerobes). To obtain this theoretical maximum yield, glucose must be fully converted to acetate as the terminal end product. In summary, the process in strict anaerobes, such as *Clostridium* sp., consists of the conversion of pyruvate to acetyl CoA and CO_2_ through PFOR, and electrons are donated to the hydrogenases via reduced FDX. This results in a maximum yield of 2 mol of H_2_ per mol of glucose. Two additional moles of H_2_ can be produced from the NADH produced during glycolysis via NADH:ferredoxin oxidoreductase (NFOR) which can donate electron to the FDX hydrogenase system, making an overall theoretical maximum yield of 4 mol of H_2_ per mol of glucose for this kind of bacteria. In facultative anaerobes, because a maximum of two molecules of formate are produced from two pyruvate molecules, the theoretical maximum yield for the PFL pathway is 2 mol of H_2_ per mol of glucose [48,49,50,51,91]. However, different constraints make the actual yields of dark fermentation much reduced. Two of the main drawbacks of dark H_2_ production are a) the existence of other fermentative competing pathways that lower the yield and b) the excessive accumulation of fermentative end products, especially acetic acid, which impairs microbial growth and H_2_ production [48,49,50,51,91]. Numerous studies have focused on the manipulation of *Clostridium* spp. and *E. coli* to enhance the H_2_ production by redirecting the fermentative pathways and reducing the accumulation of some undesired end products such as lactate, succinate or butyrate. However, the accumulation of acetic acid cannot be avoided since, in both pathways, this compound is directly linked to the production of H_2_, and its production is crucial to maintain an optimal energy/redox balance for the cells [48,49,50,51].

In order to solve the problematic acetic acid accumulation, integrative strategies combining dark bacteria and non-sulfur photosynthetic bacteria have been assessed. In these bacteria consortia, the organic acids generated by the dark bacteria can feed the photosynthetic bacteria for H_2_ production, resulting in increased H_2_ production yields (Figure 2) [47]. Theoretically, maximum yields in these integrative cultures can be obtained if acetic acid is the only secreted end product. Two molecules of acetate can be generated from glucose, in both facultative and strict anaerobes, which can then be converted into H_2_ by the PNSP bacteria, producing, theoretically, a maximum of 8 extra mol of H_2,_ and, making the overall theoretical yield of the integrative systems 10 to 12 mol of H_2_ per mol of glucose. Again, these theoretical values are not reached since different limitations exist. Among others, the use of photosynthetic bacteria in these integrative systems often requires two-stage bioreactors due to the growth incompatibility and the removal of nitrogen, which strongly inhibits the H_2_-evolving nitrogenases [47]. 

The literature concerning the use of integrative systems has considered, almost exclusively, photosynthetic bacteria as the only partners able to use and remove the acetic acid resulting from dark fermentation. However, some microalgae can be used instead of (or with) photosynthetic bacteria (Figure 2). When co-culturing Chlamydomonas with different non-H_2_ producing bacteria in acetate-free media supplemented with sugars (glucose or mannitol), algal H_2_ production can be observed if acetic acid is excreted by the bacteria. The amount of acetic acid excreted by the bacteria directly correlates with the capacity of Chlamydomonas to produce H_2_ [56]. Moreover, as demonstrated by Fakhimi et al. [56] using *E. coli* and Chlamydomonas co-cultures incubated with glucose as the sole carbon source, it is possible to produce H_2_ in a synergetic way (60% more H_2_ than the sum of the respective control monocultures), with acetic acid probably being the metabolite linking dark H_2_ production with H_2_ photoproduction (Figure 2). This study entails a proof-of-concept linking dark bacteria and algae H_2_ production. Nonetheless, the H_2_ production yield obtained in *E. coli*–Chlamydomonas co-cultures was very low and optimizations are required. 

As mentioned before, acetic acid is the only compound that Chlamydomonas can uptake as the sole carbon source for heterotrophic growth. Note that in, Chlamydomonas monocultures, no other source of organic carbon (e.g., glucose) can be used for growth or to trigger H_2_ production (Figure 2). Apart from growth promotion, acetic acid plays a significant role in H_2_ production in this alga. The presence of acetate in the medium promotes O_2_ consumption, represses CO_2_ fixation, and decreases the photosynthetic rates [92,93,94,95]; all of these factors favor H_2_ production. In addition, the presence of acetic acid in the culture media has been reported as a key parameter for photo-H_2_ production in Chlamydomonas monocultures [14] and co-cultures [56], whose role is partially independent of its capacity to promote hypoxia [14]. It has been suggested that, under light, nutrient-repleted conditions and hypoxia, the assimilation (or photoassimilation) of acetic acid, and not starch mobilization, can provide, directly or indirectly, electrons for the PSII-independent H_2_ production pathway [13,14]. Physiologically, the photoassimilation of acetate under hypoxia could be equivalent to the H_2_ photo-fermentation described in photosynthetic bacteria. 

Overall, the use of microalgae such as Chlamydomonas leads to photo-H_2_ production, while helping to bypass the drawbacks of the acetic acid accumulation and pH acidification that prevent bacterial H_2_ production. The use of algae instead of photosynthetic bacteria or cyanobacteria has the advantage of avoiding the concomitant occurrence of H_2_ uptake and the nitrogen removal from the medium, which is required to induce nitrogenases. Moreover, compared with PNSP bacteria, algae have more compatible growth conditions with some dark bacteria. Moreover, algae, but not photosynthetic bacteria, can provide extra acetate-independent H_2_ production via direct H_2_ production (PSII-dependent pathway) or via the mobilization of the starch reserves (PSII-independent pathway). The two latter pathways can potentially surpass the theoretical maximum H_2_ yield of 10–12 mol H_2_ per mol of glucose in the solely bacterial integrative systems. However, more research is still needed to explore the potential of algae–bacteria co-cocultures for H_2_ production, and to better understand how the acetate metabolism is linked to H_2_ production in Chlamydomonas anaerobic cultures. 

#### 3.1.4. Co-Culturing Chlamydomonas with Bacteria Can Alleviate the Negative Effect of S Deprivation while Promoting H_2_ Production

S deprivation is a strategy widely used to enhance photobiological-H_2_ production in Chlamydomonas [35,36], which can lead to the highest H_2_ yields (Table 1). However, this strategy has several drawbacks, including growth inhibition and the loss of the cell viability (caused by the S prolonged deficiency), which reduce the potential for H_2_ generation. Previous studies have partially overcome the harmful effects of S deprivation using continuous or semi-continuous regimes of cultivation [96,97,98]. Recently, different studies using batch co-cultures in TAP-S [58,59,65] have obtained similar results to these previous studies, although avoiding the use of continuous or semi-continuous strategies, which can greatly simplify the overall process. For example, co-culturing Chlamydomonas with *Pseudomonas* sp. [59] or with *Bradyrhizobium japonicum* [58] in TAP-S can slow the reduction in chlorophyll, enhance starch accumulation, and maintain protein content, while favoring algal H_2_ production relative to algal monocultures. However, the precise reasons why these bacteria prolonged the viability of Chlamydomonas cells in TAP-S is uncertain. Interestingly, when Chlamydomonas is incubated with the sulfur-oxidizing bacterium *Thuomonas intermedia* [65], a considerable increase in H_2_ production and algal growth are observed; these effects are even more pronounced when the cultures are treated with the oxygen scavenger Na_2_S_2_O_3_ (Table 1). Authors propose that *T. intermedia* is able to oxidize S_2_O_3_^2−^ to SO_4_^2−^, providing a S source for the alga to satisfy the minimum requirement for algal growth and, at the same time, maintain the S-deprived environment required for H_2_ photoproduction [65]. Overall, co-cultures in TAP-S require less energy inputs than continuous or semi-continuous alga monocultures and, more importantly, can support algae growth and H_2_ production simultaneously.

#### 3.1.5. Starch-Enriched Alga Biomass Can Be Used as Substrate for H_2_ Producing Bacteria

Besides the direct supply of excreted fermentative metabolites to H_2_-producing bacteria by living algal cultures, algal biomass can also support H_2_ production by strict or facultative anaerobic bacteria. Different bacteria consortia have been probed to produce H_2_ from Chlamydomonas biomass. These consortia are often composed of a fermentative bacterium and a photosynthetic bacterium. The fermentative bacteria can degrade the Chlamydomonas biomass and excrete organic acids such as ethanol, formate, acetate, propionate and butyrate, which can be used by the photosynthetic bacteria to photoproduce H_2_ via photo-fermentation. For instance, *Lactobacillus amylovorus* is able to hydrolize starch from algae biomass to lactic acid, which can feed the photo-H_2_ production in *Rhodobacter sphaeroides*, *Rhodobacter capsulata*, *Rhodospirillum rubrum* and *Rhodobium marinum* [99,100]. Similarly, *Vibrio fluvialis* converted starch accumulated in Chlamydomonas to acetic acid and ethanol, which drove H_2_ production in *Rhodobium marinum* under high salt condition [101]. Likewise, *Rhodobacter sphaeroides* produced H_2_ from formate, acetate and butyrate secreted by *Clostridium butyricum* after anaerobic fermentation of Chlamydomonas biomass [102]. In this example, direct H_2_ production from *Clostridium butyricum* fed with Chlamydomonas biomass was also attained, which can illustrate the potential of producing H_2_ from algal biomass using bacteria consortia and two-step processes (Figure 1).

## 4. Net O_2_ Evolution

In Chlamydomonas, the photoproduction of H_2_ is unavoidably linked to the photosynthetic electron chain and thereby to O_2_ generation. As mentioned before, O_2_ is a strong inhibitor of both the expression and activity of the Chlamydomonas HYDAs [103]. The measurements of O_2_ in co-cultures can be indistinctly done in the liquid phase as dissolved O_2_ (DO_2_), or in the headspace. The DO_2_ measurements are more accurate to predict HYDAs activity in Chlamydomonas co-cultures, as demonstrated by Ban et al. [59]. Nevertheless, a good correlation between these two O_2_ indices and their relationship with H_2_ production has been observed in Chlamydomonas co-cultures [60]. 

In algae monocultures, the net O_2_ evolution is a result of the O_2_ inputs and outputs. The O_2_ inputs include initial O_2_ in the headspace, the DO_2_ in the culture media, and photosynthetic O_2_ generation. The light intensity directly influences the activity of the photosynthesis processes and thereby O_2_ generation. The O_2_ outputs are due to the respiratory activity. Chlamydomonas respiratory activity greatly increases when growing heterotrophically (or mixotrophically) in acetate-containing media due to the capability of this alga to use acetate as carbon source. This is the reason why most publications use either TAP or TAP-S to study H_2_ production in this alga. 

A very simple relationship between O_2_ evolution and H_2_ production in H_2_-producing acetate-containing cultures is shown in Figure 3. In algal monocultures incubated in TAP medium, the O_2_ level quickly drops during the first 24 h. Under moderate light intensities (<50 PPFD), the photosynthetic O_2_ evolution is lower than the O_2_ consumption and the cultures remain under hypoxia for a few days, while the H_2_ production starts within the first 24 h. The hypoxic condition is maintained as far as acetic acid remains in the media. Once the acetic acid is fully consumed, the O_2_ levels rise and H_2_ production stops. Light intensity directly impacts the acetic acid uptake: the higher the light intensity, the faster the acetic acid uptake and the shorter the H_2_ production phase. At higher light intensities (>50 PPFD), there is a net positive O_2_ evolution and cultures do not reach hypoxia. [14] (Figure 3A). In co-cultures, the O_2_ outputs can be significantly increased if aerobic or facultative anaerobic bacteria are incubated in media containing organic carbon sources, which can greatly benefit H_2_ production. Again, most of the studies about H_2_ production using Chlamydomonas co-cultures are done in TAP or TAP-S media. In TAP co-cultures, the respiration rate can increase from 18% to 64% relative to Chlamydomonas monocultures, depending on the algal strain and the bacterial partners [57,61,66]. Unlike algal monocultures, no net O_2_ evolution is obtained in TAP co-cultures under moderate to high light intensity (50–100 PPFD), which allows H_2_ production at these light intensities [57,60,61,63]. A direct correlation between the presence of acetic acid in the media and the capacity to produce H_2_ have been observed in different Chlamydomonas–bacteria cultures [56,60,63,64]. Recently, Fakhimi et al. [63] have shown that the positive effect of the bacterial partners on H_2_ production can be linked to a decrease in acetate assimilation by the alga. Slower acetic acid uptake allows for a longer presence of this compound in the TAP culture medium, which, in turn, results in longer hypoxia and H_2_ production phases. This effect also allows for the use of higher light intensities compatible with H_2_ production. Distinct bacteria partners can impact the acetic acid uptake rates of Chlamydomonas differently; out of the different bacteria tested, *Pseudomonas* sp. showed the highest capacity to decrease the acetic acid uptake. All these data reveal that the use of co-cultures in TAP medium can help to reach hypoxia at higher light intensities than in monocultures, and they can increase the H_2_ yield by the means of more sustained H_2_ production. 

On the other hand, S deficiency causes a decline in PSII activity and thereby in photosynthetic O_2_ evolution [104]. In Chlamydomonas monocultures incubated in TAP-S, at light intensities above 50 PPFD, there are 1–3 days where the cultures remain aerobic (termed as lag or oxic phase). Afterwards, an anerobic phase starts and H_2_ is produced; the H_2_ production yield in TAP-S is often higher than in TAP. In TAP-S cultures, the acetic acid is never fully consumed, and its level is neither linked to the aerobic or anaerobic phases nor to the H_2_ production phase [36,105] (Figure 3B). Dark incubation prior illumination or purging with noble gases are often used as strategies to quickly deplete O_2_ levels in the TAP-S cultures and shorten the lag phase. In co-cultures incubated in TAP-S, the respiration rate is enhanced by three to eight times during the first day compared with algal monocultures, depending on the light intensity [57,59,61,66,67]. Lakatos et al. [60] observed that after just 4h of illumination, the O_2_ level in the co-cultures (4–5%) were lower than in monocultures (15–16%). These observations demonstrated that, in the case of TAP-S cultures, the co-incubation with bacteria can reduce the lag phase and avoid the dark incubation or purging required to reach hypoxia and initiate H_2_ production. 

Overall, elevating the O_2_ consumption rate by bacteria can improve H_2_ production by a) allowing the implementation of hypoxic conditions compatible with H_2_ production, b) decreasing the time required to establish hypoxia, c) extending the duration of the hypoxia phase, which directly influences the production phase, and d) tolerating higher light intensities without impairing the hypoxic conditions [59,60,63]. 

Finally, among the important aspects to be considered when setting up algae–bacteria co-cultures are the initial cell number ratios, which are one of the main concerns of many studies [57,59,61,65]. Different ratios can impact the O_2_ inputs and outputs and thereby the net O_2_ concentration in the cultures. Moreover, due to the light shading effect of the bacteria, the initial algae–bacteria ratios and light intensities should be considered and optimized at the same time [67]. According to Ban et al. [59], there is an optimum initial cell number of algae which results in the highest H_2_ production.

## 5. Extension of the Solar Spectrum Absorption Range

An important aspect of the association between microalgae and photosynthetic bacteria is the possibility to increase the range of the solar spectrum for conversion to H_2_. Microalgae and cyanobacteria can capture the visible portion of sunlight (400–700 nm) and generate H_2_, while PNSP bacteria can also capture near-infrared emissions (700–1010 nm) to produce H_2_. Therefore, an integrated system can lead to a better solar irradiation utilization. However, few studies have been carried in this sense using Chlamydomonas. Following this idea, Melis and Melnicki [106] studied a consortium of Chlamydomonas with *Rhodospirillum rubrum* to improve biomass generation. However, the light irradiance performance of this co-culture was weakly analyzed and H_2_ production was not reported for this co-culture. It would be interesting to perform a more thorough investigation of the light irradiance efficiency in similar co-cultures and their suitability for H_2_ production. 

## 6. Final Remarks

H_2_ production by microalgae is being studied due for its potential to provide a clean and renewable biofuel. However, this technology is still far from industrial application due to its low rates and yields, which make it economically unviable. In the context of improving bio-H_2_ production, strategies based on algae–bacteria consortia are still poorly explored; however, they show great potential and could be some of the best strategies to improve H_2_ production. Indeed, despite the limited number of publications, the combination of Chlamydomonas with different non-H_2_ producing bacteria is already among the most successful strategies to attain H_2_ production in this alga (Table 2). However, the future of algae–bacteria consortia remains in their capacity to integrate co-cultures with other successful strategies such as physiological treatments (e.g., S or Mg deprivation), O_2_ scavengers, cell immobilization or light modulation. Importantly, co-cultures using genetically modified strains of both algae and bacteria could also offer great potential to further improve H_2_ production.

Improved H_2_ production in Chlamydomonas co-cultures can be explained by multiple factors, including an increase in the starch content, a decline in net O_2_ evolution, a decrease in the algal acetic acid uptake, metabolite exchanges, and the utilization of higher light intensities compatible with H_2_ production. However, there are still many questions that remain uncertain regarding how non-H_2_ producing bacteria promote algal H_2_ production. 

In any case, the use of integrative systems combining different H_2_-producing microorganisms (alga, cyanobacteria, PNS bacteria and heterotrophic bacteria) could be the real challenge in the bio-H_2_ field. Combining fermentative, photofermentative and photosynthetic pathways for H_2_ production could be the most feasible approach to overcome the low bio-H_2_ production yields and make them compatible with industrial applications. In the case of microalgae, this is a very promising approach that needs to be further explored and extensively improved. A few studies have already confirmed the possibility to achieve collaborative [62,90] and even synergetic H_2_ production [56] when using Chlamydomonas together with different kinds of H_2_-producing microorganisms. This prospect can provide a new perspective on how to produce H_2_ from cheap raw materials or waste, taking advantage of microbial metabolic collaborations, while, at the same time, bypassing some H_2_ production barriers identified in both algae and bacteria (e.g., O_2_ withdrawal, acetic acid accumulation, pH control, or organic carbon and other nutrient supplies). 

However, H_2_-producing microorganisms have complex and very versatile metabolisms. Unravelling the metabolic and physiological relationships that they develop in natural ecosystems is the key to creating properly designed strategies to improve H_2_ production when co-culturing. Finding the appropriate algal and bacterial partners, suitable raw materials, and culture conditions could be the next challenge to address efficient and sustainable H_2_ production.

## Figures and Tables

**Figure 1 cells-09-01353-f001:**
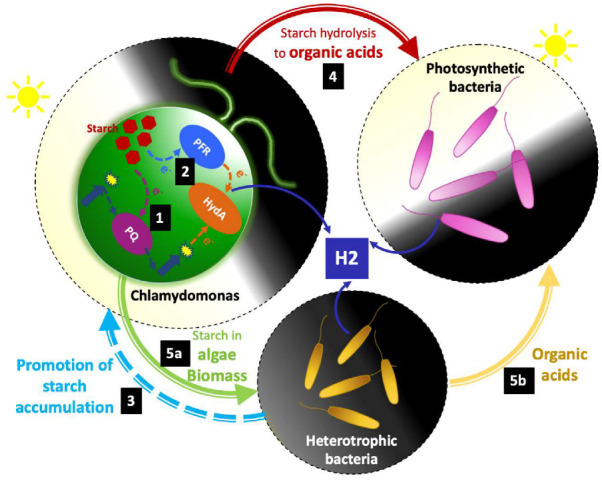
Potential starch-derived relationships between Chlamydomonas and other microorganisms during H_2_ production. Starch accumulated in Chlamydomonas cells can be used to feed the PII-independent (**1**) and fermentative (**2**) pathways. The accumulation of starch in Chlamydomonas can be favored when co-cultured with some bacterial strains (**3**). Different end products derived from Chlamydomonas starch mobilization can be excreted and used by Purple Non-Sulfur Photosynthetic (PNSP) bacteria for H_2_ production (**4**). Starch-enriched Chlamydomonas biomass can be used directly by some heterotrophic bacteria to produce H_2_ (**5a**) or in collaboration with PNSP bacteria (**5b**). Pyruvate Ferredoxin Reductase (PFR); PlastoQuinone (PQ); hydrogenase A (HydA).

**Figure 2 cells-09-01353-f002:**
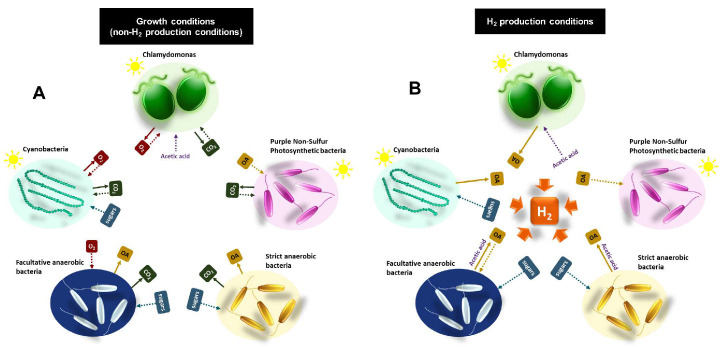
Potential metabolites exchanged among different H_2_-producing microorganisms during growth conditions (**A**) and H_2_-producing conditions (**B**). The secretion and uptake of metabolites are indicated with plain and dotted arrows, respectively. Depending on the specific culture conditions the same metabolites can be secreted or accumulated. Organic Acids (OAs) mainly include ethanol, glycerol, formate, acetic acid, lactate, succinate and butyrate. When predominant, the specific OA is indicated next to the arrow.

**Figure 3 cells-09-01353-f003:**
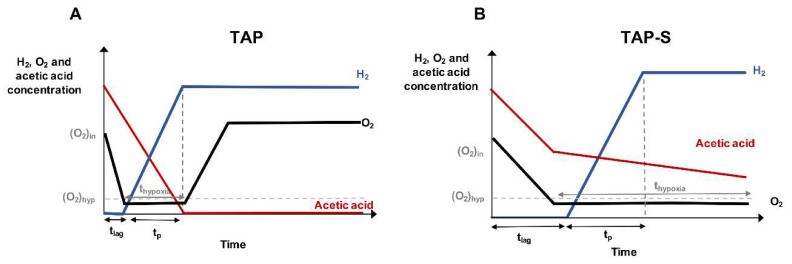
Typical trends of H_2_, O_2_ and acetic acid concentrations of Chlamydomonas cultures incubated in TAP (**A**) and TAP-S (**B**). In TAP cultures (**A**), H_2_ production occurs only in the presence of acetic acid, which is necessary to establish hypoxic conditions. In TAP-S cultures (**B**), the H_2_ production phase and the hypoxic phase are independent of the acetic acid concentration. Under the same light conditions, TAP cultures show faster acetic acid uptake and shorter lag phase than in TAP-S. H_2_ production yield and duration in TAP-S cultures is often higher than in TAP cultures. T_lag_, lag phase before H_2_ production; t_p_, H_2_ production phase; t_hypoxia_, hypoxia/anaerobic phase; (O_2_)_in_, initial O_2_ levels; (O_2_)_hyp_, minimal O_2_ levels compatible with H_2_ production.

**Table 1 cells-09-01353-t001:** Comparison of yield, rate and sustainability of H_2_ generation in Chlamydomonas–bacteria co-cultures versus alga monocultures. For each report, only data from co-cultures with their corresponding control monocultures are considered (when possible). Data are ranked according to the total H_2_ production in co-cultures.

Chlamydomonas Strain^1^	Bacterium Strain	Medium	Light Intensity (PPFD)^2^	H_2_ Production in Algal Monocultures		H_2_ Production in Co-Cultures			References
				Reported	Estimated (mL/L)^3,4^	Estimated (mL/L)^3,4,5^	Duration^5^	Estimated Average Rate (mL/L∙d)^3,4,5^	
Transgenic lba (based on cc849)	*Bradyrhizobium japonicum*	TAP-S	60	20.02 (µmol/40 mL)	≈11.22	≈170.5 (× 15.2)	14 d (× 1)	≈11.95 (× 14.9)	[57]
cc503	*B. japonicum*	TAP-S	200	70 (µmol/mg chl)	≈13.14	≈141.2 (× 10.7)	≈16 d (× 1.8)	≈8.82 (× 6)	[58]
FACHB-265	*Pseudomonas* sp. strain D	TAP-S	50	≈10 (mL/L)	10	≈130 (× 13)	≈12 d (× 3)	≈10.82 (× 4.3)	[59]
FACHB-265	*Escherichia coli* and *Pseudomonas* sp. strain D	TAP-S	50	≈20 (mL/L)	20	≈125 (× 6.2)	≈16 d (× 2)	≈7.81 (× 3.1)	[59]
FACHB-265	*Bacillus subtilis* and *Pseudomonas* sp. strain D	TAP-S	50	≈20 (mL/L)	20	≈110 (× 5.5)	≈16 d (× 2)	≈6.87 (× 2.7)	[59]
Transgenic *hemHc-lbac* (based on cc849)	*B. japonicum*	TAP-S	30	99 (µmol/mg chl)	≈21.19	≈93.2 (× 4.4)	≈16 d (× 2)	≈5.82 (× 2.2)	[58]
cc124	*B. japonicum*	TAP-S	200	20 (µmol/mg chl)	≈2.43	≈78.4 (× 32.3)	≈13 d (× 1.3)	≈6.03 (× 24.8)	[58]
FACHB-265	*Pseudomonas* sp. strain C	TAP-S	50	≈10 (mL/L)	10	≈65 (× 6.5)	≈6 d (× 1.5)	≈10.83 (× 4.3)	[59]
cc124	*E. coli (ΔhypF)*	TAP-S	50	25 (mL/L)	25	≈47.3 (× 1.9)	7 d (× 1)	≈6.75 (× 1.9)	[60]
cc849	*B. japonicum*	TAP-S	60	12.76 (µmol/40 mL)	≈7.15	≈46.5 (× 6.5)	≈8 d (× 2)	≈5.82 (× 3.2)	[57]
FACHB-265	*Herbaspirillum* sp.	TAP-S	50	≈10 (mL/L)	10	≈40 (× 4)	≈8 d (× 2)	≈5 (× 2)	[59]
cc849	*Pseudomonas* sp.	TAP-S	50	15.11 (µmol/40 mL)	≈8.46	≈34.7 (× 4.1)	≈8 d (× 2)	≈4.3 (× 2)	[61]
C238	*Rhodosprillum rubrum*	MBM	200 W/m^2^ 12:12 h L–D	0.6 (µmol/mg dry wt)	≈8.6	≈34.3 (× 4)	12 h (× 1)	≈68.54 (4)	[62]
cc849	*Stenotrophomonas* sp.	TAP-S	60	15.11 (µmol/40 mL)	≈8.46	≈33.8 (× 4)	≈6 d (× 1.5)	≈5.64 (× 2.6)	[61]
704	*E. coli*	TAP+glu^6^	12	9.7 (mL/L)	9.7	32.7 (× 3.4)	9 d (× 3)	≈3.6 (× 1.1)	[56]
FACHB-265	*Pseudomonas* sp. strain A	TAP-S	50	≈10 (mL/L)	≈10	≈30 (× 3)	≈10 d (× 4)	≈3 (× 1.2)	[59]
FACHB-265	*Phyllobacterium* sp.	TAP-S	50	≈10 (mL/L)	≈10	≈30 (× 3)	≈12 d (× 3)	≈2.5 (× 1)	[59]
FACHB-265	*E. coli*	TAP-S	50	≈20 (mL/L)	≈20	≈30 (× 1.5)	≈12 d (× 1.5)	≈2.5 (× 1)	[59]
704	*P. putida 12264*	TAP	12	17.9 (mL/L)	17.9	27.6 (× 1.5)	4 d (× 1.3)	≈6.86 (× 1.1)	[63]
704	*E. coli (ΔhypF)*	TAP+glu^6^	50	2.5 (mL/L)	2.5	26.2 (× 10.5)	4 d (× 2)	≈6.5 (× 5.2)	[56]
FACHB-265	*Bacillus subtilis*	TAP-S	50	≈20 (mL/L)	≈20	≈25 (× 1.2)	≈12 d (× 1.5)	≈2.08 (× 0.8)	[59]
cc849	*Microbacterium* sp.	TAP-S	60	15.11 (µmol/40 mL)	≈8.46	≈24.5 (× 2.9)	≈6 d (× 1.5)	≈4.09 (× 1.9)	[61]
704	*P. putida 12264*	TAP+glu^6^	50	2.5 (mL/L)	2.5	29.2 (× 11.7)	9 d (× 4.5)	≈3.2 (× 2.6)	[56]
704	*P. putida 291*	TAP	12	17.9 (mL/L)	17.9	23.1 (× 1.3)	3 d (× 1)	≈7.7 (× 1.3)	[63]
704	*P. stutzeri*	TAP	12	17.9 (mL/L)	17.9	23.1 (× 1.3)	4 d (× 1.3)	≈5.79 (× 1)	[63]
cc124	*E. coli (ΔhypF)*	TAP	50	NP	--	≈18.7 (^7^)	1 d (^7^)	≈18.67 (^7^)	[64]
704	*P. putida 12264*	TAP	100	0.8 (mL/L)	0.8	18.2 (× 22.7)	2 d (× 2)	≈9.1 (× 11.4)	[63]
704	*Rhizobium etli*	TAP	12	17.9 (mL/L)	17.9	17.7 (× 1)	3 d (× 1)	≈5.91 (× 1)	[63]
704	*E. coli*	TAP	12	17.9 (mL/L)	17.9	17.5 (× 1)	3 d (× 1)	≈5.85 (× 1)	[63]
704	*P. stutzeri*	TAP	50	4.3 (mL/L)	4.3	15.5 (× 3.6)	2 d (× 2)	≈7.74 (× 1.8)	[63]
FACHB-265	*Comamonas* sp.	TAP-S	50	≈10 (mL/L)	≈10	≈15 (× 1.5)	≈8 d (× 2)	≈1.87 (× 0.7)	[59]
704	*P. putida 12264*	TAP	50	4.3 (mL/L)	4.3	14.2 (× 3.3)	3 d (× 3)	≈4.73 (× 1.1)	[63]
cc503	*Thuomonas intermedia*	TAP-S + Na_2_S_2_O_3_	200 14:10 h L–D	43 (µmol/mg chl)	≈0.77	≈12.8 (× 16.6)	17 d (× 1.9)	≈0.75 (× 8.7)	[65]
704	*R. etli*	TAP+man^6^	50	2.5 (mL/L)	2.5	13.5 (× 5.4)	8 d (× 4)	≈1.7 (× 1.4)	[56]
704	*P. putida 291*	TAP	50	4.3 (mL/L)	4.3	10.3 (× 2.4)	3 d (× 3)	≈3.44 (× 0.8)	[63]
704	*P. stutzeri*	TAP	100	0.8 (mL/L)	0.8	8.3 (× 10.4)	1 d (× 1)	≈8.3 (× 10.4)	[63]
704	*E. coli*	TAP	50	4.3 (mL/L)	4.3	6.9 (× 1.6)	2 d (× 2)	≈3.44 (× 0.8)	[63]
*Chlamydomonas* sp.	*E. coli (ΔhypF)*	TAP	50	NP	--	≈6.8 (^7^)	1 d (^7^)	≈6.84 (^7^)	[65]
*Chlamydomonas* sp.	*Rhodococcus* sp.	TAP	Dark	≈5.6 (mL/L)	≈5.6	≈6 (× 1.1)	4 d (× 1)	≈1.5 (× 1.1)	[60]
cc124	*E. coli (ΔhypF)*	TAP	50	NP	--	5.8 (^7^)	≈22 h (^7^)	≈6.3 (^7^)	[60]
704	*R. etli*	TAP	50	4.3 (mL/L)	4.3	5.6 (× 1.3)	1 d (× 1)	≈5.6 (× 1.3)	[63]
704	*P. putida 291*	TAP	100	0.8 (mL/L)	0.8	3.5 (× 4.4)	1 d (× 1)	≈3.5 (× 4.4)	[63]
cc503	*T. intermedia*	TAP-S	200 14:10 h L–D	43 (µmol/mg chl)	≈0.8	≈3.4 (× 4.4)	17 d (× 1.9)	≈0.2 (× 2.3)	[65]
cc549	*E. coli (ΔhypF)*	TAP-S	50	0.2 (mL/L)	0.2	≈2.6 (× 13.6)	3 d (× 1.5)	≈0.9 (× 8.8)	[60]
*Chlamydomonas* sp. & *Scenedesmus* sp.	*E. coli (ΔhypF)*	TAP	50	0 (mL/L)	0	1.5 (^7^)	≈10 h (^7^)		[66]
cc549	*E. coli (ΔhypF)*	TAP	50	0	0	1.2 (^7^)	≈22 h (^7^)	≈1.3 (^7^)	[60]
*Chlamydomonas* sp. & *Scenedesmus* sp.	*Bacteria flora*	TAP	50	0 (mL/L)	0	1.1 (^7^)	≈12 h (^7^)	≈2.3 (^7^)	[66]
704	*R. etli*	TAP	100	0.8 (mL/L)	0.8	0.8 (1)	1 d (× 1)	≈0.8 (× 1)	[63]
704	*E. coli*	TAP	100	0.8 (mL/L)	0.8	0.8 (1)	1 d (× 1)	≈0.8 (× 1)	[63]
cc849	*Azotobacter chroococcum*	TAP-S	30	19 (µmol/mg chl)	--^8^	(× 3.8)^9^	≈12 d (× 1.5)	--	[67]
cc849	*A. chroococcum*	TAP-S	100	19 (µmol/mg chl)	--^8^	(× 3.6)^9^	≈8 d (× 1)	--	[67]
cc849	*A. chroococcum*	TAP-S	200	28 (µmol/mg chl)	--^8^	(× 5.3)^9^	≈10 d (× 1)	--	[67]
*Chlamydomonas* sp.	*Ralstonia eutropha*	TAP	NP	NP	--	≈1.2 (^7^)	≈1 d (^7^)	≈1.2 (^7^)	[60]
*Chlamydomonas* sp.	*R. eutropha* (Δ*hypF1F2*)	TAP	NP	NP	--	≈1.2 (^7^)	≈1 d (^7^)	≈1.2 (^7^)	[60]

^1^*Chlamydomonas reinhardtii* unless otherwise stated; ^2^ photosynthetic photon flux density (PPFD) (µmol photons · m_2_^−1^ · s^−1^); ^3^ Avogadro’s law for ideal gas is considered to estimate H_2_ productivity in the unit of (mL/L culture) 1 mole H_2_ gas (at pressure = 101.325 kPa and temperature = 273.15 K), equal to 22.41 liters of H_2_; ^4^ the average of the lowest and the highest chlorophyll concentration was considered to estimate the H_2_ productivity from “per mg chlorophyll” to “per liter culture”; ^5^ enhancements in co-cultures compared with monocultures are presented as fold changes in parentheses; ^6^ sugar is added when acetic acid is depleted in the culture media; ^7^ folds cannot be calculated because either H_2_ production in alga monocultures are zero or are not reported; ^8^ data for chlorophyll concentration was not reported; ^9^ reported fold change; Modified Bristol Medium (MBM); information not provided in the original report (NP); glucose (glu); mannitol (man); light–dark cycles (L–D); “**≈**”: data estimated from the original study (rounded values).

**Table 2 cells-09-01353-t002:** Maximum H_2_ productivity achieved by Chlamydomonas using different approaches. Data are ranked according to the total H_2_ production yield. For each study, only the maximum reported values are considered.

Strategy	Parental Alga Strain	Mutant Strain	Conditions	Reported H_2_ Production	Estimated H_2_ Production (mL/L)^1,2^	Estimated Average H_2_ Production rate (mL/L∙d)	Reference
Monoculture/Genetic modification/S deprivation	cc124	pgrl5	TAP-S, 60 PPFD	850 mL/L (9 days)	850	≈94.4	[69]
Monoculture/Genetic modification/S deprivation	cc1618	stm6	TAP-S, 100 PPFD	540 mL/L (14 days)	540	≈38.6	[70]
Monoculture/Genetic modification/S deprivation	11/32b	L159I-N230Y	TAP-S, 70 PPFD	504 mL/L (12 days)	504	≈42	[71]
Monoculture/Genetic modification/S deprivation	137c(cc124)	pgrl1	TAP-S, 200 PPFD	≈1.5 mmol/mg chl (≈5 days)	≈437	≈87.4	[34]
Monoculture/Genetic modification/S deprivation	cc1618	Stm6Glc401	TAP-S + 1 mM glucose, 450 PPDF	361 mL/L (≈8 days)	361	≈46	[72]
Consortia/*Pseudomonas* sp. /S deprivation	FACHB-265	--	TAP-S, 200 PPFD	170.8 mL/L (13 days)	170.8	13.1	[59]
Consortia/*Bradirizhobium japonicum*/S deprivation	cc849	Transgenic lba strain	TAP-S, 60 PPFD	298.54 µmol/40 mL (14 days)	≈170.5	≈11.95	[57]
Consortia/*Bradirizhobium japonicum*/S deprivation	cc503	--	TAP-S, 200 PPFD	310 µmol/mg chl (16 days)	≈164.9	≈10.3	[58]
Monoculture/S deprivation	137c (cc125)	--	TAP-S	≈155 mL/L (≈4 days)	≈155	≈38.75	[36]
Monoculture/Mg deprivation	137c (cc125)	--	TAP-Mg, 80 PPFD	6.3 mmol/L (≈8 days)	≈141.1	≈16.9	[73]
Monoculture/S deprivation/acetate free	UTEX 90 (cc1010)	--	T(A)P-S^3^, 50 PPFD, N_2_ purging	118 mL/L (4.5 days)	118	26.2	[74]
Monoculture/O_2_ scavenging	cc503	--	TAP + NaHSO_3,_ 200 PPFD	≈150 µmol/30mL (3 days)	≈112.05	≈37.3	[75]
Monoculture/Genetic modification	cc849	*hemHc-lbac*	TAP-S, N_2_ purging, dark incubation, 50 PPFD	3.3 mL/40 mL (≈5 days)	82.5	≈16.5	[76]
Monoculture/Light modulation	cc124/cc4533	--	TAP, 1 s light pulses (180 PPFD) + 9 s dark periods under Argon atmosphere	3.26 mmol/L (2.25 days)	≈73.06	≈32.5	[29]
Monoculture/acetic acid supplementation/Light modulation	704	--	TAP + acetic acid supplementation, daily aeration, 12 PPFD	65 mL/L (9 days)	65	≈10	[14]
Consortia/E. coli (*hypF*)/S deprivation	cc124	--	TAP-S, 50 PPFD	47.2 mL/L (7 days)	47.2	6.75	[60]

^1^ Avogadro’s law for ideal gas is considered to estimate H_2_ productivity in the unit of (mL/L culture): 1 mole H_2_ gas (at pressure= 101.325 kPa and temperature=273.15 K) is equal to 22.41 liters of H_2_; ^2^ the average of the lowest and the highest chlorophyll concentration was considered to estimate the H_2_ productivity from “per mg chlorophyll” to “per liter culture”; ^3^ Tris–Acetate–Phosphate (TAP) without acetate and sulfur (T(A)P-S); “**≈**”: Data estimated from the original study (rounded values); photosynthetic photon flux density (PPFD) (µmol photons · m_2_^−1^ · s^−1^).

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
