# Peer review of "Algae-Bacteria Consortia as a Strategy to Enhance H2 Production"

_cells, 2020, doi:10.3390/cells9061353_

Round 1
Reviewer 1 Report
This MN has well organized for the review of hydrogen production via algae-bacteria consortia as a strategy. It has a scientific contribution to hydrogen production research. This MN is almost ready to be accepted to publish. Only the full name of the abbreviations in the figures should be shown in the titles.
Author Response
Reviewer 1, quoted "Only the full name of the abbreviations in the figures should be shown in the titles."
Answer: The new version of the manuscript includes new abbreviations in figures and tables ‘legends as follows:
Table 1. “glu, glucose; man, mannitol; L-D, light-dark cycles;”
Figure 1: PFR, Pyruvate Ferredoxin Reductase; PQ, PlastoQuinone; HydA, hydrogenase 1
All the changes are included and tracked in the new version of the manuscript
Reviewer 2 Report
Manuscript ID: cells-791185
Title: Algae-bacteria consortia as a strategy to enhance H2 production
The current review article written and discussed well. Abstract is very clear and supports key objectives and achievements of current review. The review discussed the H2 production pathways very well with the appropriate figures. However, this review is out of scope this MDPI-Cell Journal and not provides significant contribution in the scope of MDPI-Cell Journal. This review article should be submitted in appropriate MDPI journal to get proper citation and significant contribution in the area of Algal refinery for biofuel production.
Author Response
Answer:
The subject of this review was properly notified to the Editors. A tentative title and an outline of the review content was sent several months ago.
Reviewer 3 Report
The manuscript entitled “Algae-bacteria consortia as a strategy to enhance H2 production” by Fakhimi et al., exploits the use of microalgae such as Chlamydomonas reinhardtii for the biological production of H2, nowadays the most sustainable biofuel. Different factors limit the yield of the hydrogen produced by these microorganisms also co-cultured with various photosynthetic or heterotrophic bacteria.
The manuscript describes the mechanisms leading to the production of biological hydrogen by different microorganisms and reports the achievements obtained by Chlamydomonas-bacteria consortia.
The review is well organized and provides a complete overview on the subject and a detailed analysis of the factors that could improve the H2 production.
In my opinion, the manuscript can be accepted for publication after few minor revisions:
- Page 1 and line 16, Abstract: please change in “Here, we review the different consortia that have been successfully employed….”;
- Page 2 and line 16, Section 1.1: please correct “..Among them there are the low light convertion efficiency….”;
- Page 3 and line 127, Section 2: please correct “Table 1 provide a comparative analysis of all the previously …”;
- Page 18, Caption to Figure 3: please change in “….of acetic acid necessary to establish…..”
Author Response
Answer:
- the 3 first suggestions are accepted (pages 1, 2 and 3)
- For the four suggestion (page 18), we have rephrased the sentence as follows: “In TAP cultures (A), H2 production occurs only in the presence of acetic acid, which is necessary to establish hypoxic conditions”
Round 2
Reviewer 2 Report
If the subject of this review was properly discussed with the Editors than it can be accepted for publication.